# Advances in Amine-Surface Functionalization of Inorganic Adsorbents for Water Treatment and Antimicrobial Activities: A Review

**DOI:** 10.3390/polym14030378

**Published:** 2022-01-19

**Authors:** Nabil Bouazizi, Julien Vieillard, Brahim Samir, Franck Le Derf

**Affiliations:** The Normandie Universite, UNIROUEN, INSA Rouen, CNRS, COBRA (UMR 6014), 27000 Evreux, France; julien.vieillard@univ-rouen.fr (J.V.); brahim.samir@univ-rouen.fr (B.S.); franck.lederf@univ-rouen.fr (F.L.D.)

**Keywords:** inorganic adsorbent, surface functionalization, water treatment, amine grafting, toxic molecules, antibacterial

## Abstract

In the last decade, adsorption has exhibited promising and effective outcomes as a treatment technique for wastewater contaminated with many types of pollutants such as heavy metals, dyes, pharmaceuticals, and bacteria. To achieve such effectiveness, a number of potential adsorbents have been synthesized and applied for water remediation and antimicrobial activities. Among these inorganic adsorbents (INAD), activated carbon, silica, metal oxide, metal nanoparticles, metal–organic fibers, and graphene oxide have been evaluated. In recent years, significant efforts have been made in the development of highly efficient adsorbent materials for gas and liquid phases. For gas capture and water decontamination, the most popular and known functionalization strategy is the chemical grafting of amine, due to its low cost, ecofriendliness, and effectiveness. In this context, various amines such as 3-aminopropyltriethoxysilane (APTES), diethanolamine (DEA), dendrimer-based polyamidoamine (PAMAM), branched polyethyleneimine (PEI), and others are employed for the surface modification of INADs to constitute a large panel of resource and low-cost materials usable as an alternative to conventional treatments aimed at removing organic and inorganic pollutants and pathogenic bacteria. Amine-grafted INAD has long been considered as a promising approach for the adsorption of both inorganic and organic pollutants. The goal of this review is to provide an overview of surface modifications through amine grafting and their adsorption behavior under diverse conditions. Amine grafting strategies are investigated in terms of the effects of the solvent, temperature, and the concentration precursor. The literature survey presented in this work provides evidence of the significant potential of amine-grafted INAD to remove not only various contaminants separately from polluted water, but also to remove pollutant mixtures and bacteria.

## 1. Introduction

The ever-increasing manufacturing industry and the huge range of resulting hazardous pollutants have considerably decreased the available amounts of drinking water [1,2,3]. This can have a dangerous effect not only on biological activities but also on human safety. For example, owing to incomplete use and washing operations, textile and other industries often discharge harmful dye effluents into water systems. Serious environmental pollution problems are caused by the high amount of heavy metals released in soil and water. In addition, the pharmaceutical industry widely discharges non-ecofriendly chemicals that may remain in spent water, which results in polluted effluents [4]. Toxic dyes in drinking water also pose a life threat to humans [5,6]. To solve these problems, several studies have focused on water treatment technologies [7,8,9]. Researchers recently focused on adsorption techniques to reduce water pollution [10,11,12]. Adsorption is commonly one of the best methods for reducing or removing hazardous pollutants and transforming them into safe inorganic compounds such as aminophenol [13,14,15,16]. In this context, many adsorbents have been applied to adsorb contaminants from water, e.g., metal oxide nanoparticles (MOx), activated carbon, biomass, graphene oxide, textile, polymers, clay, and many other sophisticated porous materials [7,8,9,10,11,12,13,14,15,16]. According to their main composition, the sophisticated porous materials can be classified as metal oxides/hydroxides, carbon-based materials, organic polymers, fibrous materials, and agricultural waste. Figure 1 presents a view of the most effective and recently employed adsorbents for removing contaminants from liquid solution.

Despite the interesting properties and results on water treatment, the majority of these adsorbents showed a visible weakening in the continuous removal of toxic pollutants, poor reusability, and low adsorption capacity due to their surface properties. However, inorganic adsorbents (INADs) have attracted more interest in comparison to the other categories of adsorbents. This was explained by their surface properties and adsorptive characters: INADs lose their efficiency on water treatment because they are not stable enough. In this regard, techniques for surface modification and activation have been envisaged as effective pathways to enhance the separation and the effectiveness of INAD adsorbents [17,18,19,20,21,22,23]. Many works have been conducted to functionalize the porous materials and add more activated sites at the adsorbent surface. In order to improve the surface properties of these INADs, numerous organic molecules have been used to add new activated sites at the INAD surface [19,20,21,22,23]. Synthetic and natural organic compounds have demonstrated many advantages onto various materials, i.e., a simpler preparation technique, a lower cost, and effective strategies for the preparation of advanced composites. Among these organic molecules, amines are considered as the smart molecules with high values because they can increase both the adsorptive and antibacterial properties. Both natural and synthetic amines have a great effect on INAD stability and a great efficiency during environmental application. More particularly, 3-aminopropyltriethoxysilane (APTES), diethanolamine (DEA), dendrimer-based polyamidoamine PAMAM, and branched polyethyleneimine (PEI) are extensively employed for INAD functionalization and applications in microbiological and environmental catalysis, gas capture, and medical applications (Figure 2) [17,18,19,20,21,22,23]. INAD functionalization via the chemical grafting of amines has increased the number of useful activated sites for adsorption and capture of toxic molecules. Metal oxides based on copper oxide (CuO), zinc oxide (ZnO), and tin oxide (SnO_2_) modified via chemical grafting of amines possess superior electrical, optical, catalytic, and antibacterial properties not attainable via their metallic alloy and monometallic counterparts [17,21,24,25,26].

Chemical grafting of amine at the INAD surface occurs via a covalent interaction in which the molecules can persist for a long time. Covalent grafting of amine is one of the most sustainable and successful methods; it slightly decreases the surface area, while it increases the adsorption capacity. This method is regarded as a more efficient way of altering the specific properties of INAD and as a straightforward modification approach while significantly improving INAD properties. The existence of a positively charged amine group facilitates the attachment of many hazardous pollutants such as negatively charged bacterial cells, negative toxic molecules, and cationic and anionic dyes in liquid solution [27,28,29,30,31,32]. In other works, immobilized amine has been used to design and manufacture highly advanced materials such as 2D and 3D composites. Amine is employed as a link between the core and the shell for the synthesis of core–shell materials where amine plays a key role in the stabilization and sustainability properties of the resulting materials. The enhanced stability and surface compatibility of amine with porous materials can be achieved through the kinetic trapping of amines surrounding the porous materials. However, further efforts are still required to fully elucidate the intimate relationship between amine-modified INAD structures and their adsorption of wastewater pollutants and their antibacterial properties [19,20,21,22,23,24,25,26,27,28,29,30,31,32]. The goal of this review is to provide a view of the surface functionalization via grafting of amines and their adsorption behavior in water treatment. The strategies utilized for the grafting of amine at the INAD surface are established and compared in terms of concentration, temperature, and solvent. In addition, the reactions and interactions are investigated. The literature survey presented in this paper provides evidence of the good potential of amine-grafted INAD materials for removing several contaminants from aqueous solution through adsorption. The contents of this review are illustrated in Figure 3.

## 2. Amine for Surface Functionalization of Adsorbents

The surface properties of INAD play an important role on the extent of the interaction between adsorbents and adsorbates. Consequently, surface modification and functionalization have been targeted to improve the stability and adsorptive properties of porous materials for water treatment and to increase their adsorption capacity, as well as reducing toxicity. INAD adsorbents generally have a negatively charged surface coming from the hydroxyl groups (-OH) and saturated oxygens. As a result, INADs become unstable and their efficiency can be highly reduced. Consequently, in most cases, INADs lose their adsorptive properties. This can be explained by the competitive interaction between the negatively charged elements (i.e., the OH groups and unsaturated oxygen) and the adsorbates [18,33,34,35]. INAD based on metal-loaded biomass and hydrochar has been functionalized by APTES grafting to lower the number of OH groups present at the INAD surface. Hydroxyl groups have been found necessary for the grafting of high amounts of amine onto the INAD surface (Figure 4). By comparing the adsorption capacity of an unmodified adsorbent with a negatively charged surface and its modified counterpart, it is clear that hydroxyl groups destabilize the materials involving a competition between adsorbents/adsorbate. The adsorption capacity of unmodified adsorbents is lower than that of their modified counterparts. Amine-modified adsorbents visibly increase the adsorption uptake of various toxic molecules and improve the reutilization of the materials, suggesting their high stability. Despite its low adsorption efficiency, it is important to note that negative surface charge can be advantageously used for subsequent surface modifications via covalent coupling using silane chemistry or via physical adsorption or electrostatic interactions.

Surface functionalization of INADs can be carried out by distinct strategies involving both chemical and physical interactions. For amine grafting, the relevant interactions likely involved during the reaction process are covalent interactions, hydrogen bonding, and electrostatic interactions. Most of these interactions enhance the surface properties, particularly surface activation and the adsorptive character. Covalent functionalization at the INAD surface based on silica is made possible by the well-established silane chemistry, which mediates strong linkages between the INAD surface and aminosilane. Despite being well established, silane chemistry can present drawbacks that result in steric hindrance and/or uneven reactivity depending on the coupling agents, and limitation to molecules with reactive groups for the covalent linking step. However, hydrogen bonding is widely used for adsorbing gas molecules, which is beneficial for the reversible reaction between gas release and adsorption. For example, the grafting of amine based on APTES at the metal-loaded biochar and metal oxide demonstrated that amine could involve interactions between the gas molecules and aminosilane via both hydrogen bonding and electrostatic interactions, as supported in Figure 5. 

Until now, hydrogen bonding and electrostatic interactions have been the most commonly used strategies for the preparation and functionalization of adsorbents for reversible gas adsorption. However, amine grafting via covalent interactions has played a key role in the improvement of adsorptive properties and recyclability, particularly for wastewater treatment. Table 1 reports different amine-modified INAD adsorbents employed for removing various pollutants from water. Bouazizi et al. employed several types of amine such as APTES, PAMAM, DEA, and others for the surface functionalization of porous materials [18,20,22,24]. Results obtained in water treatment showed that covalent grafting of amine onto the adsorbent is of great interest to increase the adsorption uptake and the number of recycling cycles. Most of these amine molecules have been commonly used as stabilizing and coupling agents because they promote strong covalent linkages onto the INAD. For example, surface functionalization with PAMAM has been extensively used and is another example of such a type of surface modification. PAMAM with terminal NH_2_ groups is a dendrimer with amine terminal groups that acts as a protective layer around the adsorbent surface in these situations. This method has quite often been used to attach microorganisms such as Gram+ and Gram− bacterial strains *Staphylococcus epidermidis* and *Escherichia coli* and organic pollutants (nitrophenols and dyes). Due to its opposite charges, PAMAM interacts with the surface of colloidal nanoparticles [18,19]. While these grafting methods are efficient for the adsorption application, the grafting strategies depend on the solvent, the concentration, and the temperature of the media.

## 3. Effect of the Solvent

The solvent is an essential parameter for the grafting of amines at the surface of INADs, and can have a great influence on the effectiveness of the reaction and the rate of grafted amines (Table 2). The solvent affects the surface and the wetting properties of the material, and this favors the control of the grafting process at the surface or the inner material. Researchers recently studied the functionalization of mineral-clay-based INADs by using 3-aminopropyltriethoxysilane (APTES) in the presence of various solvents such as distilled water, tetrahydrofuran, toluene, and ethylene glycol. The solvents with a low surface energy wetted the adsorbent easily and thereby made it possible for silane to interact with the -OH groups present at the INAD surface. However, the wetting process was lower for the solvents with a higher surface energy for water, and this relatively decreased surface adsorption. In other works, ethylene glycol was used as a solvent for chemical grafting of amines which indicated a low amount of aminosilane incorporated onto the INAD [62]. When organic solvents such as cyclohexane and toluene are used as anhydrous products, the evaluation of amine grafting at the INAD surface takes a long time (20 h) for the 3-aminopropyltetraethoxysilane using the reflux method [63]. The obtained adsorbent shows a visible change of the structure materials, with an expanded or distorted molecular structure (Table 2). In summary, the presence of water facilitates the initial hydrolysis of amines on the INAD surface, depending on the volume of water [64,65]. Results of the degradation of toxic pollutants by various INADs reported in literature show that APTES grafting has a superior adsorption capacity of toxic molecules in the presence of H_2_O/amine as compared with the anhydrous or hydrated solvent. In detail, when water is absent from the synthesis protocol, amines bind directly to the hydroxyl group at the surface of the INAD. However, excess water in the medium promotes the polymerization of amines during the functionalization process. 

The nature of the dispersant medium has been pointed out too: protic solvents are not favorable to the grafting process because of potential competition reactions. A competitive interaction is indeed involved between the alkyl siloxane and hydroxyl groups of the solvent, and it allows H-bonding rather than hydroxyl group bonding. Aprotic solvent has been used to enhance favorable solvent–surface interactions [66,67]. To explain the mechanism occurring during the grafting onto silica-based INAD, the following steps are the main reactions involved in the process. Firstly, amine enters a hydrogen bonding interaction: the hydroxyl groups at the surface or the basic amine function inter-react with a proton from a hydroxyl group and produce an ionic bond. Due to the last type of interaction, which is more stable than the first one, the hydrogen-bonded molecules self-catalyze the condensation on the silicon side of the silane molecule, and a covalent bond is formed. Upon condensation on the silicon side, the amine group loses its interaction with the surface and the amine points away from the surface. The higher number of ethoxy groups on the APTES molecule leads to a much faster stabilization. Thus, the aminosilane molecule turns from the original amine-down position to an amine-up position. 

## 4. Effect of Temperature

An increase or decrease in the synthesis temperature can influence the resulting products, but also play a crucial role in the amine grafting at the INAD surface. Various synthesis temperatures have been used for grafting amines onto INAD (Table 2). In most cases, the temperature rise is typically going along with an increase in the interactions between the amine molecules and the surface hydroxyl groups. Xue et al. [65,79] studied the role of temperature on the grafting of APTES in the presence of toluene as a solvent for a series of temperatures (15 °C, 30 °C, 45 °C, 60 °C, and 75 °C). The results showed that chemical grafting of APTES was promoted by increased temperature. Bouazizi et al. and Bertuoli [17,35,80] investigated the grafting of APTES at the INAD surface in ethanol/water (75:25, *v*/*v*) at 50 °C and 80 °C. The temperature increase allowed for a high interaction of the amine groups with the adsorbent. High temperature (220 °C) was also found to be the favorite pathway for the diffusion of amines onto the adsorbent [81]. Based on the above findings, elevating the concentration of amines such as 3-aminopropylphosphonic acid (3APPA) or 3-propylphosphonic acid (3PPA) and high temperature results in an increased modification degree.

## 5. Effect of the Concentration

In order to ensure effective grafting of amines at the INAD surface, the amine concentration plays a crucial role in the preparation of advanced functional adsorbent materials. Considering the weight of amines, high amounts of amine can affect the properties of the support used for post treatment. On the one hand, increases in the amine concentration can mean that more active terminal groups such as -NH_2_ have interacted with the adsorbent surface via -OH groups. In this way, the high amount of amine can increase the terminal target of the functional groups on INAD as a support. Xue et al. [65] investigated the grafting of aminosilane with different ratios, and the results showed an improvement of the adsorption capacity for dye removal when the ratio is increased. On the other hand, in this context, the adsorption uptake decreased slightly when the amine concentration increased. The high number of amine molecules that surrounded the INAD surface affected its porosity and decreased its adsorption capacity. The enhancement of the adsorption uptake can be explained by the incorporation of high numbers of amines onto the INAD surface. Until now, no explanation has been found as to why the adsorption capacity of INAD decreases when the concentration of the precursors increases. In other works, a large molecular weight of amines improved the stability of the adsorbent [82,83]. PEI functionalized with 1,2-epoxybutane (EB) was synthesized by Choi et al. [83]. The proportion of primary amine gradually decreased, while the proportion of secondary amine and tertiary amine gradually increased. Consequently, the grafting of 1,2-epoxybutane onto INAD improved the stability and the adsorptive properties. More recently, they also demonstrated that increases in the molecular weight of amines induced superior thermal stability. The resulting material had O/N molar ratios of 0.42, 0.64, and 0.82, respectively, as shown in Figure 1. 

## 6. Influence of Amines on the Hydrophilic Characters

The grafting of amines can influence the hydrophilicity and hydrophobicity of INADs. In order to investigate the effect of amine immobilization, contact angle measurements were employed to study wettability, surface energy, and diffusion resistance. Bouazizi et al. investigated the hydrophilic properties of INADs modified with amines such as DEA, PAMAM, and aminosilane. The contact angle results showed that the INAD surface became more hydrophilic than the original one. Amine intercalation in the adsorbents increased their hydrophilic character, and this improved diffusion or the transfer rate of organic molecules from the aqueous solution toward the INAD surface [18]. In addition, amine grafting introduced a visible decay of the hydrophilic surface, evidenced by the loss of the OH stretching bond [19]. This behavior was also recorded for DEA(OH)_2_ grafting, which decreased the hydrophilic character, given the consecutive decrease until total loss of the hydroxyl groups. Therefore, we can conclude that amine grafting increases the hydrophilic character as it decreases the number of hydroxyl groups present at the adsorbent surface. In another work, amine grafting induced a visible decrease in the hydrophilic character in relation to slight compaction of the structure. This can be explained by hydrophobic interactions within the aminosilane entanglement and strong interactions between amino and surrounding -OH groups. In this regard, these interactions imply that diffusion is hindered, causing slow initial wettability without affecting the hydrophilic character. On the contrary, metal–organic frameworks (MOF) functionalized by inserting 2-aminoterephthalic acid showed an increase in the hydrophobic character of the adsorbent [77]. The surface morphology of both unmodified and amine-modified MOF remained unchanged, but their hydrophilic properties decreased, as supported in Figure 2.

It was recently demonstrated that the contact angle increased as the number of amine groups decreased. While the quantity of amines rose from 0.1 vol.% to 10 vol.%, the contact angle increased, suggesting a higher coverage of the INAD surface. This last result is of great interest for wastewater treatment because it shows that amine molecules can interact with the adsorbent surface in a well-organized manner to maximize their adsorptive properties. Again, the hydrophilic characteristics of amine-modified adsorbents are explained by the remaining hydroxyl groups that lower the contact angle. Consequently, the obtained hydrophilic charters are very helpful for removing pollutants from water. 

## 7. Chemical State of the Amine Groups

FTIR and XPS analysis were employed to investigate the presence of the different chemical states of the amine groups and the surface compatibilities attributed to those states. The band observed at 1582 cm^−1^ was associated to the bending vibration of NH_2_, which is visible as a shoulder on the deformation mode of molecular adsorbed water (Figure 3) [85,86]. The broad band between 1540 and 1490 cm^−1^ was associated to the asymmetric deformation vibration of the NH_3_^+^ groups [87,88,89,90]. In addition, the position of the NH_2_ band shifted toward the lower wavenumbers as compared with those recorded for the aqueous solutions of primary amines. This was explained by the presence of hydrogen bonding interactions with NH_2_ groups [91]. In this case, the nitrogen atom (N) is considered as a hydrogen acceptor [87]. The same trend was obtained by Bouazizi et al., where a similar shift (from 1600 to about 1575 cm^−1^) was recorded for APTES grafted with metal oxide and activated carbon [23]. These results evidence an interaction between amine and silanol groups, which result in a variety of hydrogen-bonded surface conformations such as an intramolecular membered ring structure. Another confirmation was obtained by the presence of hydroxyl groups (Ti–OH) available for hydrogen bonding and or acid/base interactions. Additionally, amine grafting onto INAD induced distortion of the surface chemistry resulting in structure expansion in some cases and structure compaction in others.

XPS analysis provided data about the amine molecules through high-resolution observation of N1s. The peak observed at 399.8 eV was attributed to the traces of molecularly adsorbed N_2_. However, upon amine addition, the N1s peak of amine-loaded INAD consisted of a broad asymmetric peak composed of two bands at 398 and 400 eV (Figure 4). Until now, it has been hard to draw a conclusion about these peaks, as N1s data showed various types of interactions that can have a similar binding energy, which are considered as complicating factors [93,94,95]. The peaks at low binding energy (398 eV) can be associated to the non-interacting NH_2_ groups or to Lewis acid-base interactions. Consequently, the component at high binding energy at 401 eV is associated to protonated amine (NH_3_^+^) groups originating from proton transfer from -OH groups. The hydrogen-bonded amine groups are hard to observe experimentally by XPS. Although other peaks’ position and ratio of NH_2_ and NH_3_^+^ for INAD evidenced that amines addition occurred with specific surface arrangements. [92]. Basically, the above results prove that possible surface conformations of amines such as 3APPA at the INAD surface can be established via the free NH_2_ groups accessible for interactions with their surroundings. In other cases, the NH_2_ groups can involve intra- and inter-adsorbate interactions and adsorbate–surface interactions, with 3APPA amine as the adsorbate [95]. 

Recent research has been conducted in the field of amine utilization for biomedical applications. Amines have proved very useful for an effective junction and design of high-value biomedical INADs [96]. In this case, amines play a key role to elaborate a biomedical hydrogel in 2D and 3D shapes. In this case, amines allow for the formation of links or connections between other chemicals at the surface to ensure the creation of surface compatibility. Amines can adhere to and design the surface materials via several interaction types and relate bond chemistry to the emergent adhesive properties with a specific emphasis on biomedical applications (Figure 6).

## 8. Role of Amines in the Stability of Adsorbents

Various types of amines have been widely employed to generate activated sites and improve the surface properties of INADs for environmental applications such as water treatment. Moreover, several authors have proposed that a high stability of the adsorbent is of great importance as it increases the efficiency of toxicant removal and reusability. In this aim, researchers estimated that addition of organic moieties bearing chelating groups on the support was effective to avoid the non-stabilization and aggregation of adsorbent particles [98]. Among these organic moieties, amines have attracted many researchers for the elaboration and functionalization of INADs through the chemical grafting of amines and/or its derivates. In this regard, many works have shown that amine surface functionalization increases INAD stability and the number of reusability cycles. Figure 5 and Figure 6 show an example of the mechanisms taking place during the process of APTES-functionalized graphene oxide (GO) [24]. APTES plays a crucial role in INAD stabilization based on graphene oxide as an adsorbent and it increases the immobilization yield of metallic nanoparticles. The results obtained with APTES-modified GO showed not only increases in the adsorption uptake toward organic pollutants but also improved reusability of the catalyst.

The same trend was obtained for palladium stabilization on amine-functionalized zeolite. In other words, INAD materials can be stabilized by using amine as the organic surface in the organic–inorganic composite. Infrared spectroscopy measurements showed a marked change upon the incorporation of amines. This was observed by the slight decay in band intensity noticed for the 3400 cm^−1^ and 2919 cm^−1^ peaks associated to the asymmetric stretching vibrations of NH_2_ and the C–H aliphatic groups. The authors explained these results by the significant role of amine insertion in the molecular structure of the INAD, which induced a compaction of the organic entanglement due to the occurrence of strong interactions between metal and amino groups M:NH_2_ [99].

The above phenomenon is also reported in the literature for INAD based on metal nanoparticles, where the CH and OH groups involved interactions with NH_2_ resulting in INAD stabilization [101]. These results clearly demonstrate that intercalation of amine-like APTES contributes to adsorbent stabilization due to strong interactions between the adsorbent and NH_2_. Importantly, the thermal analysis demonstrated that untreated INAD has a low thermal stability, which is considered a drawback for many applications. However, amine addition played an interesting role by enhancing thermal stability. This was illustrated by the visible decrease in mass weight above 100 °C. Thus, chemical grafting of APTES increased the thermal stability of INAD, and induced more stable oxygen-functionalized groups, resulting in a higher temperature of around 500 °C to decompose the silanol groups or Si–O–Si bonds during amine surface functionalization of INAD. APTES and PAMAM as amine sources appear to be responsible for the thermal stability of the grafted materials. Concerning INAD durability, amines have a potential effect on INAD reusability. The results obtained on the removal of pollutants onto metal oxide, activated carbon, and fibrous materials revealed that immobilization of amine at the INAD surface had an interesting role as to INAD reusability and recyclability. Measurements of the catalytic activities of INAD showed that a copper oxide catalyst modified by the chemical grafting of amines could be recycled and reused several times for water treatment. This type of INAD was used more than seven times for 4-NP reduction, eight cycles for the removal of methylene blue (MB), five cycles for the elimination of malachite green (MG), and eight cycles for the removal of remazol red (RR) without visible decay in its catalytic capacity. This superior removal uptake is explained by the key role of APTES amine, which acts as an effective shield that prevents leaching and promotes material durability, and by the high stability of the catalysts (Figure 7). This was confirmed by the structure and thermal stability after the catalytic process. Both FT-IR and TGA analyses revealed no visible change compared with the unmodified counterpart (Figure 8).

## 9. Pollutant Removal by Amine-Grafted Inorganic Adsorbents

Tremendous amounts of minerals and organic pollutants can be found in water. This is why amine-modified INADs have attracted many scientists because they are effective for treating polluted waters and wastewater. This section studies the removal of various pollutants and bacteria using amine-grafted INADs. 

### 9.1. Removal of Heavy Metals and Nitrates

Table 3 summarizes the adsorption capacity of various INADs functionalized by amine grafting for removing dyes, nitrates, heavy metals, organic molecules, and pharmaceuticals from polluted water. The discussion starts with the removal of heavy metals and nitrate, which is a monovalent anion. Consequently, it allows positively charged surfaces or surfaces containing exchangeable anions to be effective adsorption sites. Liu et al. demonstrated that nitrate adsorption involved different kinds of adsorption processes such as electrostatic interactions, cation exchange, complexation, and hydrogen bonding [102]. These interactions depend on the composition, structure, and surface properties of the INAD and the adsorption conditions (Figure 7).

The chemical functionalization of an INAD using aluminum-loaded activated carbon with grafting of the APTES for fluoride removal was studied by Bakhta et al. [103] and the sorption capacity of fluoride onto the functionalized INAD reached 92.0 mg g^−1^. Fotsing et al. [104] prepared cocoa shell biomass via chemical grafting of APTES and PEI. This functionalized INAD was more effective for removing Cr(VI) and nitrates, with an approximate adsorption uptake of 100 mg g^−1^. Furthermore, INADs showed affinity toward both nitrates and heavy metals as an interesting property of theses adsorbents. This was explained by the role of amine immobilization at the INAD surface. In addition, the sorption of nitrate and Cr(VI) onto amine-modified biomass as a function of the pH improved from pH 2 to 5, with a maximum experimental adsorption capacity of 16.71 mg g^−1^ at pH 5. These results explained the competition between the amine and nitrate groups due to the interference of silane present at the INAD surface. However, at higher pH values, competition between OH^−^ and NO_3_^−^ may have occurred, resulting in decreased NO_3_^−^ adsorption, and causing electrostatic repulsion between the surface and NO_3−_ ions. Furthermore, Bao et al. [56] studied the removal of Zn(II) by functionalized Fe_3_O_4_@SiO_2_ with amine groups. INADs were found to exhibit a sorption capacity around 169.5 mg g^−1^ at pH 5. Dindar et al. [105] modified INAD based on mesoporous silicate with a solution of APTES, and the final products were used to eliminate the Cr(VI), As(V), and Hg(II) ions. This investigation showed that the sorption efficiency depends on the number of amine groups present on each INAD surface. The APTES-modified INAD exhibited a sorption uptake around 47 mg g^−1^, while the INAD treated with N-[3-trimethoxysilyl-propyl] ethylenediamine had a maximum sorption capacity around 140 mg g^−1^ at pH 1.7. The presence of amine groups at the INAD surface greatly improved the adsorption uptake of hazardous heavy metals and nitrates. In addition, they found out that the loading of more amine groups enhanced the adsorption capacity, as in the materials prepared with ethylenediaminepropyle salicylaldimine. In another study, modification of natural bentonite by anchoring APTES and 3,2-aminoethylaminopropyltrimetoxysilane (AEAPS) was prepared and applied to remove Pb(II) in aqueous solution [106]. Marjanović et al. [107] functionalized natural and acid-activated sepiolites by grafting, using the [3-(2-aminoethylamino)propyl] trimethoxysilane precursor. The material was used to remove chromium (VI) from aqueous solution. Different adsorption mechanisms can occur during heavy metal removal. The main adsorption phenomenon for heavy metals involves electrostatic attraction and hydrogen bonds. Keshvardoostchokami et al. [108] proved that a chitosan-based amine source improved the removal of pollutants such as nitrate and ammonia. Ammonia removal involved ion exchange, whereas nitrate removal implied hydrogen bonding and electrostatic bonding (Figure 8).

### 9.2. Removal of Dyes

Regarding the high amounts of untreated dyes (methylene blue, malachite green, crystal violet, remazol red, and others) discharged by various industries into the environment, amines have been considered a potential candidate for activating INADs for effective removal of these dyes. The control of dye degradation has traditionally been studied at different initial pollutant concentrations to measure the capacity of INADs to remove these toxic pollutants. A decreased absorbance of the solution over time reflected the progressive removal of the dyes from the solutions. By comparing the time required for reaching total dye removal by CuO-based unfunctionalized and amine-functionalized INADs, no dye was removed by the “raw” INAD. Nevertheless, amine-INAD showed a total dye removal in less than 2 min contact time [20,22]. Consequently, amine-loaded INAD displayed a superior adsorption uptake in comparison with its unmodified counterpart. This result is of great importance because it provides clear evidence that amine grafting onto INAD plays a key role in catalytic activities. In addition, the results revealed the important role played by amines in the stabilization of the INAD catalysts, as explained in the above sections. Xue et al. [65] grafted amine at the attapulgite surface to be used as an INAD for removing reactive dyes in aqueous solution. Amine grafting onto the INAD presented a very high adsorption capacity reaching 99.32% for various dyes such as MB and RR. The difference in the dye adsorption capacities was explained by the electrostatic attraction between reactive dyes and the protonated grafted amino groups. Lou et al. [110] reported the synthesis of APTES-Fe_3_O_4_/bentonite material and its application for MB adsorption. The maximum adsorption uptake was around 92 mg g^−1^ as compared with its unmodified counterpart. Araghi et al. [111] reported the preparation of amino-loaded silica magnetite nanoparticles as an INAD, which illustrates the role of amine grafting in dye removal. Similarly, Morshed et al. investigated another type of INAD based on metal-fibrous materials and its modification with the PAMAM for cationic MB and MG dye elimination [112]. The adsorption capacities of the amine-grafted INAD were 49.48 mg g^−1^ and 47.03 mg g^−1^ for MB and MG, respectively. Furthermore, despite the role played by amine groups in dye removal, the porosity of the adsorbent should not be neglected during the sorption process. For cationic dyes, Laaz et al. [113] reported adsorption of red congo and anionic brilliant green onto SBA-15 and amine-functionalized SBA-15 as INADs. Red congo removal significantly depended on the number of grafted amine functional groups rather than on the porosity of the material. Therefore, increasing the number of amine groups at the INAD surface can improve the capacities for dye adsorption. Additionally, the hydrophilic character and the bond interactions between dyes and amine functional groups explain the high adsorption capacities of amino-silicate-based INADs. 

The adsorption of dyes from contaminated water onto the surface of an adsorbent can be achieved via various adsorption mechanisms [114]. Dye adsorption onto INADs involves many processes such as surface complexation, electrostatic interactions, van der Waals forces, surface diffusion, and intraparticle pore diffusion, as shown in Figure 9.

### 9.3. Elimination of Organic Pollutants

The treatment of hazardous organic compounds present in wastewaters has attracted great attention, and the control of water pollution has become one of the major challenges worldwide. Major classes of nitrophenolic molecules act as organic pollutants when they are released in water by numerous industries [115,116,117,118]. Amine-grafted INADs have shown positive results for eliminating these organic pollutants. Furthermore, methods for reducing and removing these pollutants, e.g., adsorption and catalytic reduction, have been considered the most effective techniques for water depollution. Bouazizi et al. studied the removal of 4-NP organic dye via catalytic reduction in graphene oxide (GO) and APTES-functionalized GO as INAD materials [24]. Measurements of the UV-absorption bands showed a strong decay of the 400 nm band for 4-NP and increase in the 290 nm peak for 4-AP. Importantly, this effect was even more pronounced in the presence of APTES amine. This provides evidence of the beneficial contribution of amine grafting for both catalytic reduction and nanoparticle immobilization. Catalytic performance was related to the key role of APTES grafting in enhancing the electron transfer from the INAD surface to the nitro group of 4-NP [119,120,121]. It is worth noting that the reduction yield of the organic molecules was around 97.5% in less than 2 min, suggesting that amine-functionalized catalysts can rapidly generate 4-AP as ecofriendly molecules [122,123]. As previously stated, the grafting of APTES improved the removal of toxic pollutants. The slight decrease in the concentration of 4-NP may be associated with the adsorption behavior of the 4-NP molecules onto the amino-INAD [124]. Interestingly, kinetic studies on APTES-modified INADs showed a constant rate k around 0.481 min^−1^, superior to those of unmodified INAD. These results can be explained by the existence of a strong amino/support interaction effect between the amine groups (APTES) and the INAD (GO sheets) [125]. 

The improvement of the catalytic properties was further associated to organic pollutant diffusion towards the solid surface of the INAD. Similarly, works on metal-fibrous material-based INADs and their amine-functionalized counterpart for organic element removal showed that APTES grafting was key to improving electron transfer in link with the catalytic capacity. To verify the role played by amines in the depollution of water from organic molecules, control experiments were performed to study the reduction in 4-NP by calculating the rate constant. The time course of catalysis strongly depended on the post-treatment steps, and evidenced a positive effect of amine grafting onto INAD, as supported by a higher value of k. Finally, the removal of organic molecules by INADs can occur via hydrogen bonding. More particularly, adsorption of organic contaminants with terminal NO_2_, SO_2_, and CO_2_ onto amine-modified INAD involves hydrogen bonding (Figure 10). 

### 9.4. Degradation of Mixtures of Pollutants

Mixtures of organic molecules and dyes can be released in the environment in the form of industrial effluents, and this complicates their removal during the water remediation process. In this regard, elimination of these hazardous molecules is of great importance, as it is an urgent issue. Following the same water treatment protocol as mentioned in the above sections, various amine-modified INADs have been used for removing pollutant mixtures. INADs had an affinity toward the adsorption of mixtures of organic pollutants and dyes (Figure 9). An aqueous solution containing a mixture of nitrophenol and dyes was typically employed to evaluate the adsorption capacity of INADs functionalized by APTES grafting (Table 3). The pollutants fully disappeared within a few minutes, suggesting that INADs are efficient for removing pollutant mixtures. A comparison of the removal of different organic molecules such as nitrophenol and MB dyes demonstrated that nitrophenol required more time for its complete elimination. However, these results showed that amine-functionalized INADs are more efficient for removing organic pollutants than for removing dyes. This was explained by the unavoidable protonation of 4-NP groups which could favor the adsorption of dye. 

## 10. Amines for Biomedical Applications

Day after day in the past decades, the use of antibiotics has increased and the number of resistant bacterial strains has increased too. On the path of searching for effective materials with antibacterial properties, amine-modified INADs displayed an interesting behavior toward inhibition of bacteria. The antimicrobial properties of amine-functionalized INADs were evaluated by diffusivity and inhibitory tests towards the two Gram+ and Gram− bacterial strains *Staphylococcus epidermidis* and *Escherichia coli*, respectively. Bouazizi et al. studied the antibacterial capacity of copper oxide modified by amine intercalation on the growth of *E. coli* and *S. epidermidis*. Optical density measurements showed that antibacterial activity occurred when a state of partial CuO dissolution and Cu^+^ cation release was reached. Interestingly, the addition of amine-based DEA at the CuO surface highly increased the antibacterial capacity, suggesting that amine groups may restrict the release of Cu^+^ cation [17,19,21]. Subsequently, chemical grafting of -NH- can penetrate the cell and release the cation inside, and then damage the bacterium. In other words, the protonation of amine groups is attributed to the antibacterial property, involving sufficiently strong electrostatic interactions with the negatively charged bacterial membranes, and resulting in inhibited bacterial growth [17,21]. Therefore, antibacterial activity was improved, evidencing the key role of amine functionalization for CuO nanosheets. In the same vein, results obtained with CuO-Si-S-NH_2_ showed that greater bacterial inhibition was observed after APTES grafting. Researchers explained that the number of free electrons at the nanosheet interface increased following amine addition, and made the resulting INAD more efficient in killing bacteria such as *S. epidermidis* and *E. coli.* These results were confirmed using amine-functionalized metal-fibrous materials, indicating that -NH_2_ grafting was involved in damaging the bacterial cells: optical density measurements went down to zero (Figure 10). Consequently, the chemical grafting of amines is the main parameter that endows metal oxides with antibacterial activity against both Gram+ and Gram− bacteria. The mechanism can be explained as follows: the surface area of the nanoparticles adheres to powder strands which in turn increases the contact between the powder and the bacteria. Meanwhile, the amine groups increase the junction with the powder particles, and thus the contact with the bacteria is prolonged. A longer contact time between nanoparticles and bacteria causes more damage to the bacteria so that the antibacterial capacity is improved. The antibacterial activity is also explained by the oxygen species generated by the reaction, such as H_2_O_2_ and O_2_•. More of them are produced after the intercalation of amine molecules. Therefore, this improvement is closely related to the Cu, Si, and N elements. 

Amine surface functionalization of INADs caused the rupture of bacterial cells, their shrinkage, and their death. This confirms the antibacterial effect of the amine-functionalized samples. Another work reached the same conclusion using trimethylamine (TEA)-grafted zeolite. Figure 11 summarizes the inhibition zone diameter values of untreated and INAD-treated solution. The unmodified INAD possessed antibacterial activity against both bacteria. However, when amine groups were inserted within the INAD structure, the antibacterial activity was further enhanced. Measurements of the diameter of the inhibition zone showed 27.9% and 64.9% increases for *E. coli* and *S. aureus*, respectively. This improvement in the antibacterial capacity was attributed to the synergistic effects of the positively charged protonated amine groups grafted onto the INAD. This behavior is in good agreement with the literature [131,132,133]. According to these studies, such increases in the positively charged surfaces can actually increase the interaction between the INAD and the negatively charged bacterial cells. 

Based on the morphology of bacterial cells, researchers clearly observed and controlled these antibacterial properties. In its normal state, *E. coli* appears as a rod with an intact cell-shaped structure [134]. After amine grafting onto the INAD surface, holes were observed on the bacterial membranes. Therefore, amine addition disrupted the bacterial membrane, causing leakage of the bacterial cytoplasmic materials [135]. This behavior is well documented for *E. coli* as compared with *S. aureus*. In similar conditions, the morphology of *S. aureus* did not appear to be damaged (Figure 12). This might be due to a different killing mechanism in which the damage mostly affected the proteins, lipids, or inner components of the bacteria, rather than the bacterial membrane [136]. The different mechanisms could be due to the different membranes of *E. coli* and *S. aureus*. The thicker membrane of Gram+ bacteria better protects the membrane from damage [135,136]. 

## 11. Conclusions

Various inorganic adsorbents (INADs) based on metal oxide, metal-loaded fibrous materials, graphene oxide, metal–organic frameworks, silica, and metal-loaded biomass were functionalized by amine grafting for improved removal of pollutants in wastewater and antibacterial activity. Chemical functionalization of INADs via the grafting of amine depends on many parameters such as the nature of the solvent, the temperature, the amount of water, and the quantity of amine precursors. Amines are an effective agent for INADs because they induce superior activated sites at the surface of materials, which are very suitable for various applications. Due to the key roles of amines, INADs have been successfully used for adsorbing and removing heavy metals, dyes, organic molecules, mixtures of pollutants, and bacteria from water. Amine fixation can occur via covalent interactions, hydrogen bonding, and electrostatic interactions. Further kinds of amine grafting will be very useful and can play a key role to enhance both catalytic and antibacterial activities. Amine-functionalized INADs can be employed to produce highly activated sites at the grafted surface, which can act as supplementary activated sites. Furthermore, amine-modified INADs enhance antibacterial activity due to the presence of highly reactive oxygen species. The next challenge will be to upscale the experiments towards an industrial scale. Finally, the use of amine-grafted INADs for water disinfection is expected to be explored in the future.

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
