# Peer review of "Advances in Amine-Surface Functionalization of Inorganic Adsorbents for Water Treatment and Antimicrobial Activities: A Review"

_polymers, 2022, doi:10.3390/polym14030378_

Round 1
Reviewer 1 Report
This review paper is interesting. However, the following are the comments.
(i) Redraw figure 3 with the help of software.
(ii) On page 13 of lines 301, 302, 303: "The hydrophilic characters, of amine-modified adsorbents, is explained by the remaining hydroxyl groups lower the contact angle. Consequently, the obtained hydrophilic charters are very benefit for the removal of pollutants in water" Make more clear by showing the relation between contact angles and hydrophobicity.
(iii) English correction is needed throughout the manuscript.
(iv) The reviewer wants to see the high resolution of scheme 4.
(v) Redraw figure 11.
(vi) Why are there two schemes 4( pages 16 and 27)?
(vii) Rewrite the conclusion by exact findings from the discussion of the manuscript.
(viii)reduce self-citations.
Author Response
Reviewer #1: Comments from the reviewer:
This review paper is interesting. However, the following are the comments.
(i) Redraw figure 3 with the help of software.
Author’s answer: Done, Figure 3 is replaced by another with high quality. Changes are marked in yellow color in the revised manuscript.
Comments from the reviewer:
(ii) On page 13 of lines 301, 302, 303: "The hydrophilic characters, of amine-modified adsorbents, is explained by the remaining hydroxyl groups lower the contact angle. Consequently, the obtained hydrophilic charters are very benefit for the removal of pollutants in water" Make more clear by showing the relation between contact angles and hydrophobicity.
Author’s answer: We agree with the reviewer’s comment. Indeed, measurement of contact angle over inorganic adsorbents showed that amine grafting is accompanied with a decreases of contact angle ( i.e the drop deposited was adsorbed by the sample). Consequently, the adsorbent has affinity to adsorb water. However, if the deposited drop was not adsorbed over the sample, the measured contact angle will be higher suggesting that adsorbent had a hydrophobic character. To complete with the reviewer’s comment, more detail is added in the corresponding section. Changes are marked in yellow color in the revised manuscript.Comments from the reviewer:
(iii) English correction is needed throughout the manuscript.
Author’s answer: Done, the manuscript has thoroughly corrected and all changes made accordingly have been marked in yellow in the revised manuscript.
Comments from the reviewer:(iv) The reviewer wants to see the high resolution of scheme 4.
Author’s answer : Done, Scheme 4 is replaced by another one with high resolution. Change are marked in yellow color in the revised manuscript
Comments from the reviewer:(v) Redraw figure 11.
Author’s answer : Done, as requested, figure 11 is replaced. Change are marked in yellow color in the revised manuscript
Comments from the reviewer:(vi) Why are there two schemes 4( pages 16 and 27)?
Author’s answer : We agree with the reviewer’s comment, this was an error, now all scheme and figures are re-numbered.
Comments from the reviewer:
(vii) Rewrite the conclusion by exact findings from the discussion of the manuscript.
Author’s answer : We are grateful for the reviewer’s evaluation, now the conclusion is rewritten according the finding from the discussion of the manuscript. Changes are marked in yellow color in the revised manuscript.
Comments from the reviewer:
(viii)reduce self-citations.
Author’s answer : Done, as requested, our self-citation are reduced, and all changes are marked in yellow color in the revised manuscript.
Reviewer 2 Report
The paper presents research on the advances in amine-surface functionalization of inorganic-adsorbents for water treatments and antimicrobial activities. The presentation of methods and scientific results in the current form is satisfactory for publication in Polymers journal. The paper is interesting, but unfortunately, a bit is sloppily written. For this reason, I would like to see the next version of the article again before the final acceptance. The minor and significant drawbacks to be addressed can be specified as follows:
1. “(PAMAM)NH2” (line 19) or “PAMAM” (0. Nomenclatures, page 2)? Which version is correct?
2. Page 2. (i) 3-Aminopropyl)trimethoxysilane ---> 3-aminopropyl)trimethoxysilane (ii) What does the parentheses in the name “3-aminopropyl)trimethoxysilane”? (iii) Polyamidoamine- NH2 ---> polyamidoamine- NH2 (iv) 2-Aminoterephthalic ---> 2-aminoterephthalic (v) Metal oxide ---> metal oxide (vi) Pentaethylenehexamine ---> pentaethylenehexamine (vii) Trimethylamine ---> trimethylamine. Please, check all the manuscript.
3. Fig. 1. (i) And hydroxides ---> and hydroxides (ii) Does the background colour matter? Green and yellow for Fe2O3 and blue for polyamide? Why?
4. Fig. 2. (2), i.e. 2 in the circle?
5. (i) Scheme 1 ---> Figure 3. In my opinion, Figs 1 and 2 are also schemes!!! Consequently, all figures need to be renumbered. (ii) AC and GO. Please explain AC (activated carbon) and GO (graphene oxide) in “0. Nomenclatures”. GO is explained in line 547 (iii) Other schemes should also be renumbered. In my opinion, for example, Fig. 11 is also a scheme!!!
6. Line 157. Figure captions. gas molecule ---> gas molecule (CO2).
7. Tab. 1. I observe problems with names again. Sometimes they are written with a lowercase letter and once with a capital letter (see, for example, (i) Removal or removal (ii) Chitosan or chitosan). Throughout the work, please, standardize this table (see also other ones).
8. Line 238. 3PPA?
9. Tab. 2, PAMAM (NH2)4. 4?
10. “et al” or “et al.”
11. Fig. 5, Infrared spectra of 3APPA grafted mesoporous TiO2. 20mM50? Please, give the additional details in the figure captions.
12. Fig. 6. Which sample? Are they the same as shown in Fig. 5? Can differences be found between these samples?
13. Line 361. “a c covalent”? “a c”?
14. Fig. 11. CS is Cocoa shell (CS)?
15. (i) Problem with dots, i.e. line 436: Table .3 ---> Table 3 or Tab. 3 (ii) figure. 11 ---> Figure 11
16. (i) line 351 Giovanni Bovone ---> Bovone (ii) line 441 Y. Liu ---> Liu (iii) M. Keshvardoostchokami ---> Keshvardoostchokami (iv)
17. Line 480. Red letters?
18. Fig. 13. Mechanism ---> mechanism.
19. (i) Tab. 3. mg/g ---> mg g-1 (ii) line 451 mg/g ---> mg g-1
20. There should be a chapter on using these materials on an industrial scale.
21. I lack clear recommendations on the direction in which the research should go and which material is the best and/or the most perspective. Maybe there should be a chapter - Perspectives.
22. References (i) Titles of the articles. Words starting with a lowercase or uppercase letter. (ii) [134]-[138] journal titles - italic font (iii) minor mistakes – line 1071 “S., ... & Lu, J. “ Please check and standardize all the references.
Author Response
Reviewer #2:
Comments from the reviewer:
The paper presents research on the advances in amine-surface functionalization of inorganic-adsorbents for water treatments and antimicrobial activities. The presentation of methods and scientific results in the current form is satisfactory for publication in Polymers journal. The paper is interesting, but unfortunately, a bit is sloppily written. For this reason, I would like to see the next version of the article again before the final acceptance. The minor and significant drawbacks to be addressed can be specified as follows:
1. “(PAMAM)NH2” (line 19) or “PAMAM” (0. Nomenclatures, page 2)? Which version is correct?
Author’s answer : We agree with the reviewer’s comment, this was an error, the (PAMAM)NH2 is the correct nomenclature. This was revised in the manuscript and changes are marked in yellow color.
Comments from the reviewer:
- Page 2. (i) 3-Aminopropyl)trimethoxysilane ---> 3-aminopropyl)trimethoxysilane (ii) What does the parentheses in the name “3-aminopropyl)trimethoxysilane”? (iii) Polyamidoamine- NH2 ---> polyamidoamine- NH2 (iv) 2-Aminoterephthalic ---> 2-aminoterephthalic (v) Metal oxide ---> metal oxide (vi) Pentaethylenehexamine ---> pentaethylenehexamine (vii) Trimethylamine ---> trimethylamine. Please, check all the manuscript.
Author’s answer : We agree with the reviewer’s comment, the names, abbreviations and nomenclatures are revised and the whole manuscript is checked carefully. Changes are marked in yellow color in the revised manuscript.
Comments from the reviewer:
- Fig. 1. (i) And hydroxides ---> and hydroxides (ii) Does the background colour matter? Green and yellow for Fe2O3 and blue for polyamide? Why?
Author’s answer : We are grateful for your comment and suggestion, figure 1 is replaced by another with high resolution. There is no significance of the backround color for all materias present in the figure 1. Now the background color is modified. Change are marked in yellow color n the revised manuscript.
Comments from the reviewer:
- Fig. 2. (2), i.e. 2 in the circle?
Author’s answer : We agree with the reviewer’s comment, this was an error and now the (2) is removed from the figure.
Comments from the reviewer:
- (i) Scheme 1 ---> Figure 3. In my opinion, Figs 1 and 2 are also schemes!!! Consequently, all figures need to be renumbered. (ii) AC and GO. Please explain AC (activated carbon) and GO (graphene oxide) in “0. Nomenclatures”. GO is explained in line 547 (iii) Other schemes should also be renumbered. In my opinion, for example, Fig. 11 is also a scheme!!!
Author’s answer : We agree with the reviewer’s comment, figures and schemes are renumbered accordingly. AC and GO were supplemented in the section of nomenclature. Changes are merked in yellow color in the revised manuscript.
Comments from the reviewer:
- Line 157. Figure captions. gas molecule ---> gas molecule (CO2).
Author’s answer: Done, Figure caption has been revised, as requested. Changes are marked in yellow color in the revised manuscript.
Comments from the reviewer:
- Tab. 1. I observe problems with names again. Sometimes they are written with a lowercase letter and once with a capital letter (see, for example, (i) Removal or removal (ii) Chitosan or chitosan). Throughout the work, please, standardize this table (see also other ones).
Author’s answer : Thank you for your comment, we have revised carefully the table 1 and changes are marked in yellow color in the revised manuscript.
Comments from the reviewer:
- Line 238. 3PPA?
Author’s answer : We agree with the reviewer’s comment, now the full name of 3PPA (3-propylphosphonic acid) was added in the nomenclature table. Changes are marked in yellow color in the revised manuscript.
Comments from the reviewer:
- Tab. 2, PAMAM (NH2)4. 4?
Author’s answer : We agree with the reviewer’s comment, this was an error and now the manuscript was checked carefully. Changes are marked in yellow color in the revised manuscript.
Comments from the reviewer:10. “et al” or “et al.”Author’s answer: Done, this an error (et al. and not et al) this has been revised accordingly. Changes are marked in yellow color in the revised manuscript.Comments from the reviewer:
- Fig. 5, Infrared spectra of 3APPA grafted mesoporous TiO2. 20mM50? Please, give the additional details in the figure captions.
Author’s answer: Done, more details about the infrared spectra are added in the figure caption. Changes are marked in yellow color in the revised manuscript.
Comments from the reviewer:
- Fig. 6. Which sample? Are they the same as shown in Fig. 5? Can differences be found between these samples?
Author’s answer : Thank you for your comment, figure 6 presents the XPS analysis of amine (3APPA) grafted TiO2 sample. XPS characterization showed that no differences between the samples.
Comments from the reviewer:
- Line 361. “a c covalent”? “a c”?
Author’s answer: Thank you very much for your precious remark, this was an error and now it was revised. Changes are marked in yellow color in the revised manuscript.
Comments from the reviewer:
- Fig. 11. CS is Cocoa shell (CS)?
Author’s answer : Done, CS is the Cocoa Shell (CS), now the full name has been added in the revised manuscript and changes are marked in yellow color.
Comments from the reviewer:
- (i) Problem with dots, i.e. line 436: Table .3 ---> Table 3 or Tab. 3 (ii) figure. 11 ---> Figure 11
Author’s answer : Thank you very much for your comment, now the manuscript has been revised accordingly and changes are marked in yellow color.
Comments from the reviewer:
- (i) line 351 Giovanni Bovone ---> Bovone (ii) line 441 Y. Liu ---> Liu (iii) M. Keshvardoostchokami ---> Keshvardoostchokami (iv)
Author’s answer : Donne, it has been revised accordingly. Changes are marked in yellow color in the revised manuscript.
Comments from the reviewer:
- Line 480. Red letters?
Author’s answer : Thank your for your comment, it was an error and now it was modified. Changes are marked in yellow color in the revised manuscript.
Comments from the reviewer:
- Fig. 13. Mechanism ---> mechanism.
Author’s answer : Done, this has been corrected accordingly and changes are marked in yellow color in the revised manuscript.
Comments from the reviewer:19. (i) Tab. 3. mg/g ---> mg g-1 (ii) line 451 mg/g ---> mg g-1
Author’s answer : Done, change are marked in yellow color in the revised manuscript.
Comments from the reviewer:
- There should be a chapter on using these materials on an industrial scale.
Author’s answer : Thank you for this interesting suggestion, indeed a chapter is in progress in this direction (i. e materials for chemical processes, materials for reactor and column filtration etc..)
Comments from the reviewer:
- I lack clear recommendations on the direction in which the research should go and which material is the best and/or the most perspective. Maybe there should be a chapter - Perspectives.
Author’s answer : Thank your for your valuable suggestion, a chapter is in progress in these thematic.
Comments from the reviewer:
- References (i) Titles of the articles. Words starting with a lowercase or uppercase letter. (ii) [134]-[138] journal titles - italic font (iii) minor mistakes – line 1071 “S., ... & Lu, J. “ Please check and standardize all the references.
Author’s answer : Done, now all the references are standardized and checked. Change are marke in yellow color in the revised manuscript.
Round 2
Reviewer 1 Report
The revised manuscript looks fine.
Reviewer 2 Report
The authors have addressed my queries satisfactorily. The manuscript was improved in accordance with my suggestions and I have no further objection to this paper. The quality of the paper has been improved and, therefore, I consider that the article can be accepted in its present form.